# The Effect of an Alternative Swimming Learning Program on Skills, Technique, Performance, and Salivary Cortisol Concentration at Primary School Ages Novice Swimmers

**DOI:** 10.3390/healthcare9091234

**Published:** 2021-09-19

**Authors:** Konstantinos Papadimitriou, Dimitrios Loupos

**Affiliations:** Faculty of Physical Education and Sport Science of Thessaloniki, Aristotle University of Thessaloniki, 57001 Thermi, Greece; loupo@phed.auth.gr

**Keywords:** backstroke, breaststroke, start, sink, cortisol

## Abstract

The playful training method shows positive effects on sports learning, thus the aim of the present study was to compare the effect of two different swimming learning programs. In an 8-week intervention with a training frequency of three times per week, 23 healthy primary school-aged novice swimmers (13 boys, 10 girls) aged 9.0 ± 0.9 participated. They were split into control (CG) and alternative (AG) groups and evaluated on skills (Start, Sink), backstroke (BK) and breaststroke (BR) technique, performance (Skills time, Kicks Time), and salivary cortisol concentration. According to the results, “Start” had a greater percentage of success in AG, at the first (CG = 9.1% vs. AG = 58.3%, *p* = 0.027) and third (CG = 63.6% vs. AG = 100%, *p* = 0.037) measurement. Additionally, greater scores were found in technique for AG in both BK (*p* = 0.009, *η*^2^ = 0.283) and BR (*p* = 0.020, *η*^2^ = 0.231). Salivary cortisol concentration was decreased for both groups (*p* < 0.001) and greater in CG at the second measurement (*p* < 0.001). The alternative swimming learning program was found to be more efficient or equally effective, compared with the standardized method in-water skills, swimming technique and performance, and in salivary cortisol concentration.

## 1. Introduction

The playful training method for learning a sport or a technique has been used by coaches on children at various sports and levels. The aim of playful training is to entertain the children and to make them feel spontaneous, with and without rules, and demand to succeed via their participation in the training [1]. The benefits of the method are the faster technique assimilation and pleasure that children show with their participation in a program that contains organized games for the learning of specific skills [2]. The background of this learning approach originates from the theory of constructivism, which makes learning ability more effective when the trainee participates in the understanding and enjoyment of a movement than when he remains a passive receiver. Additionally, it is highlighted that with the playful approach, exhausting, boring, and high repeatability exercises, which are contained in a classic training method and are often used by coaches, are avoided [3].

In tennis, a six-week intervention study was conducted on 62 children aged 11 years old, targeted to learn the sport through a playful learning program (Play and Stay) [1]. The sample was divided into an interventional and a control group. The interventional group used the learning skills via a normal game, whereas the control group used several exercises that coaches use for the same skills’ methodological teaching. The skills were assessed, namely service, forehand, and backhand, before the intervention, at the sixth week, and one week later from the intervention’s end. Study results showed that the 36 children who participated in the intervention group had a greater improvement in the skills that were taught compared with the 26 children who followed the classic learning [1].

Another sport in which the effect of playful training method was assessed was table tennis. In the study 56 students participated and they were divided into control and playful training groups. The target was to learn the technical elements of table tennis such as service, attack, and ball guidance. According to the results, after 48 min of training twice a week during the academic year, there was a greater improvement in attack and service in the playful training group [4].

Common results about the effectiveness of playful teaching in sports show several reviews and meta-analyses. Additionally, the authors have reported a disagreement about the using adult-oriented training on children, due to the monotony which that kind of training presents, and as a result, the children abandon their activity [5,6]. In another meta-analysis, a total of 15 studies has shown that a program which contains playful exercises positively affects the participation of children in training. Specifically, these exercises motivate them to participate in a sport which is more enjoyable. Moreover, the playful exercises contribute to the development of motor learning, game and execution skills, and decision-making in the training.

Back to the interventional studies, Blatsis [2] examined the effect of the International Amateur Athletic Federation (IAAF) Kids program on 226 children aged 11–12 years old, 102 of whom were trained in a playful manner, while the rest were trained with a classic learning method on different track and field skills. The study was conducted over a period of 12 weeks. The results revealed that the playful method was more effective than the classic one in parameters such as Yo-Yo test, long jump, and agility.

The playful teaching approach was utilized by Miller [7] too, who assessed the effect of that kind of teaching, compared with a classical method in common skills such as throwing, handling, and perceptual ability. Intervention’s duration was seven weeks with the participation of 107 students aged about 10 years old. According to the results, it was observed that playful training was more effective on performance improvement, in pleasure questionnaire scores, and perceptual ability, in contrast with the control group.

Similarly, in 40 children aged 12–13 years old it was found that the playful approach promotes the learning effectiveness of traditional Malaysian games. The duration of the intervention was eight weeks, with three training sessions of one hour per week. The content of traditional games included skills such as running, jumping, space perception, and children’s socialization, since such games do not contain any participation numerical limitation. Program effects were evaluated through speed, agility, and balance tests. The results showed that the children who used the playful approach were improved significantly in the factors that were evaluated [8].

In swimming, the only alternative learning approach, apart from the use of high repetition standardized exercises, was used for the development of the sense and perception of the forces exerted from the water on the children’s body during their movement. Thus, in swimming, the use of an alternative training approach targeting on fun and learning, such as “Play and Stay” or “IAAF Kids” programs, must be further studied [9]. According to the literature, the authors show that the target of playful method is to increase: (a) children’s will to participate, (b) fun, and (c) mood via their participation in a sport [1,2].

Cortisol concentration is a useful indicator for the examination of children’s mood [10]. In a study with 117 children, aged 3 to 6 years old, the cortisol concentration was examined in the morning and afternoon. The children were divided into three groups depending on the level of care (low, medium, or high) which was provided by each daycare center, where they were attending at least three times per week. A reduced cortisol concentration was found in the daycare centers with high-quality services. In contrast, in the children who participated in daycare centers with lower care services, a higher cortisol concentration was observed [10].

In swimming, the cortisol concentration tended to decline after an acute period in low-intensity exercise as opposed to high-intensity exercise [11]. Similarly, after acute high-intensity exercise in children aged from 9 to 10 years old, increased cortisol concentration was found [12].

Thus, because of the lack of literature about the effect of playful training in swimming, and the use of an approach that targets mimetic ability, game, and the sensation of the body in the water, the aim of the present study was to compare the effect of a classic versus an alternative swimming learning program on skills (Start, Sink), on backstroke (BK) and breaststroke (BR) technique, on swimming performance (Skills time, Kicks Time), and in salivary cortisol concentration on primary school ages novice swimmers.

## 2. Materials and Methods

### 2.1. Participants

Power analysis indicated that a sample size of 10 participants per group would be needed to detect significant differences [13]. Participants were recruited via an annual summer swimming program. Inclusion criteria were (a) healthy participants, (b) aged between 8–10 years old, (c) novices at swimming, (d) biological maturation (≤ 2) (Tanner’s scale) [14], and participation in the sport at least after their fifth year of age [15]. Exclusion criteria were (a) non the inclusion, (b) pharmaceutical treatment, (c) any disorder, or (d) participation in another sport during the intervantion.

A total of 40 swimmers participated in the intervention. Twenty-three of them completed all of the measurements and had more than 75% percent attendance, while none of them continued the follow-up measurements. The 23 novice swimmers (13 boys, 10 girls), (Median, SD) aged 9.0 ± 0.9, were recorded for their height, weight, training age, attendance, training volume, and Tanner’s scale, which is a five-point scale which defines physical measurements of development based on external primary and secondary sex characteristics. Humans’ biological maturation affects learning ability [15] (Table 1).

Then, the 23 swimmers were randomly split into two groups according to demographic characteristics (age, height, weight, Tanner’s scale, training age, (*p* > 0.05)) (Table 1), the baseline values of the Kicks time (sec) test (*p* > 0.05), and their preference. These groups were the control (CG) and alternative (AG), in which 11 and 12 swimmers participated, respectively. Before the intervention all the swimmers and their parents were informed about the study’s process and the safety of the measurements. Then, a consent form was signed by parents to ensure swimmers’ participation. The study was planned and conducted according to the Code of Research Ethics of Aristotle University of Thessaloniki.

### 2.2. Intervention’s Details

A parallel randomized design was used to compare the effects of two swimming learning programs (Standardized (CG) vs. Alternative (AG)) in-water skills, swimming technique and performance, and salivary cortisol concentration. The duration of the intervention was eight weeks [16], with three training sessions per week [17] and one day off between each session. The duration of each training session in both groups was 45’ and included 3–4 exercises. The repetitions of each exercise occurred according to the swimmers’ ability to make it successful. The intervention was conducted in an open 17.5 m pool, 27 degrees Celsius, during the summer months (June to August) at Sohos, Thessaloniki, Greece. Νo follow-up period was occurred because it was the summer holiday period for all the participants.

Groups’ training sessions comprised swimming exercises for the skills of start and sink and for the styles of backstroke and breaststroke. These skills were chosen because they are necessary for those ages. Additionally, the styles of backstroke and breaststroke were chosen because of the difference in the move’s symmetry, the multiple muscle activation, and the novelty of their use compared with freestyle, which is usually used in studies.

The main difference between the groups (CG and AG) was the approach that the skills and styles were taught. The CG followed a usual training approach which contained standardized exercises that swimming instructors use in high repetitions. On the other hand, the AG used alternative exercises named Tec Pa, in which the children had to use their imagination, and during each exercise experience from daily situations was added with the use of different kinds of objects which helped the children to make a more precise move.

The study was organized with the contribution of five swimming coaches to ensure the blinded measurements and objectivity. Thus, one coach was used to plan the swimming sessions, one to train both groups according to the plans, two more to assess the swimmers in the three measurements during the intervention, and the last one to analyze the variables by which the swimmers were assessed.

### 2.3. Alternative vs. Standardized Exercises

Table 2 and Appendix A show some of the exercises that were used between the two groups. The target of both groups was to instruct the same skills and technique but with a different approach. Tec Pa’s alternative exercises were targeting to increase the swimmers’ imaginations and the ability to find solutions during an exercise.

Specifically, the swimmers sometimes had to play the role of a soldier, fire worker, diver, etc., using different objects which helped them to precisely perform the skills or the styles’ moves. In contrast, the CG had to perform a number of exercises in which the swimmers should follow the coach’s demands without the use of their imagination or of some objects that could help the learning process.

### 2.4. Measurements

The measurements occurred before (June), in the middle (4 weeks) (July) and at the end (8 weeks) (August) of the intervention (Table 3). At the beginning of each month and in the same day, the children were evaluated for their technique (Backstroke (BK), Breaststroke (BR)), then for their swimming skills (Start, Sink 1 & 2) and performance. In the middle of the intervention (4th week), in the second training session of the week the salivary cortisol was collected before and after swimming training.

### 2.5. Technique’s Evaluation

Swimmers in both groups and in the three measurements were evaluated in BK and BR techniques by the same experienced swimming coach with the use of Tec Pa cards [18]. Tec Pa is an evaluation tool which assesses six key points of a swimming styles’ technique. These key points are the position of the head, the position of the body, the elbows, the knees, the ankles, and the full body coordination. All swimmers in each group had to swim 15 m of each style (BK and BR) with the command to swim slowly and as carefully as possible. Their unique attempts in both styles were recorded on a camera which was placed on a high spot for better evaluation from the coach. Then, one coach assessed swimmers’ technique by watching the videos and recording swimmers’ scores via Tec Pa.

### 2.6. Skills and Performance Measurement

After the evaluation of the technique, the Skills and Performance evaluation followed. The skills included “Start”, in which the swimmers had to enter the pool first with their hands and then with the other parts of the body. Then, they had to swim for 5 m as far as the pool’s rope keeping their head outside the water. When the swimmers reached the pool’s rope, they performed the first of the two dives, “Sink 1”. The swimmers had to dive under the lane’s rope without any of their body parts touching it. After the first dive they swam for another 5 m as far as the opposite side of the pool. The exact process was repeated while performing the second dive “Sink 2”, and the skills measurement was finished.

Skills were recorded as successful, with one point, or unsuccessful, with zero points. Additionally, the time that the swimmers needed to complete these Skills (Skills’ Time) was recorded. When the swimmers completed the skills, they took a kickboard and immediately continued with the kicks’ performance (Kicks’ Time), in which the swimmers covered the distance of 35 m as fast as possible. At the end of this process, the coach stopped the stopwatch (TYR Z - 100 LAP) and the swimmers had completed their try (Figure 1). To ensure the reliability of the measurements, during the evaluation of swimmers’ performance, the try of each swimmer were recorded via a digital video camera (Sony DCR-HC52 MiniDV Handycam Camcorder with 40x Optical Zoom). Additionally, the coach had an assistant coach who was observing all the processes and was noting the result of each swimmer.

The conceptualization of this skills and performance test was chosen to make the children feel that they were participating in a game instead of the usual test that several authors utilize to evaluate children’s’ performance. Both groups enjoyed that type of evaluation and this gave confidence to the children for further training participation.

### 2.7. Saliva Cortisol Concentration

The saliva cortisol concentration measurement occurred only at the fourth week of the intervention to examine the children’s mood between the two swimming learning programs. A total of 0.5 ml of saliva was collected from all the swimmers before and immediately after the swimming training. Saliva was collected in tubes that were saved in a portable fridge, then the samples were analyzed in the laboratory with the ELISA method [19].

The measurement’s validity was ensured by following the process of Hanrahan et al. [20]. Thus, the children were informed to avoid food, liquid consumption, brushing their teeth, and chewing gum 30 min before the measurement. At the end of training, five minutes before the sampling, swimmers washed their mouth with cold water.

### 2.8. Statistical Analysis

The variables’ values were shown as median with standard deviation (±). Descriptive statistic and test of normality (Shapiro–Wilk) (*p* > 0.05) for all the variables were used for a sample of fewer than 50 participants. Categorical variables of “Start” and two “Dives” (Sink 1, Sink 2) between the two groups were analyzed using Fisher’s Exact Test (χ^2^) for 2 × 2 Table. Additionally, Chochran’s Q Test was used to examine possible difference between three measurements.

Continue variables of performance and technique were analyzed with the parametric analysis of two–way ANOVA with repeated measures (group * measurements), checking for possible within or between subjects’ effects. Additionally, the measurements were checked for Homogeneity and Sphericity (*p* > 0.05). When homogeneity did not meet, the ratio of G2/G1 was checked (G2/G1 > 1.5). Additionally, Mauchly’s test and Greenhouse Geisser were used for the measurements’ sphericity. Possible statistically significant difference between subjects’ effects were analyzed via Syntax, making pairwise comparisons between groups with Bonferroni’s post hoc test.

Additionally, two–way ANOVA was used to measure cortisol concentration (groups * measurements), checking Levene’s test for homogeneity (*p* > 0.05) and possible interaction between groups with Wilk’s lambda. In all continuous variables, the Effect Size (ES) with Partial Eta square (*η*^2^) were calculated. The analysis was performed with the statistical software IBM SPSS Statistics for Windows, Version 27.0. Armonk, NY: IBM Corp. The level of significance was set at *a* = 0.05.

## 3. Results

### 3.1. Categorical Variables 

#### Start, Sink 1, and Sink 2

According to Fisher’s exact analysis, the categorical variable of “Start” had a statistically significant difference, with a greater percentage of successful tries, in AG, at the first (CG = 9.1% (1/11 participants) vs. AG = 58.3% (7/12 participants), *p* = 0.027) and third (CG = 63.6% (7/11 participants) vs. AG = 100% (12/12 participants) (*p* = 0.037)) measurement, respectively. Additionally, according to Cochran’s analysis both of “Start” (*p* = 0.001), “Sink 1” (*p* = 0.003), and “Sink 2” (*p* = 0.046) variables, there was a statistically significant difference between the three measurements (Table 4).

### 3.2. Continuous Variables

According to the normality test (Shapiro–Wilk), normality was found in a sample with fewer of 50 participants (*p* > 0.05), thus followed parametric analysis in all continuous variables. 

### 3.3. Skills Time, Kick Time, Sum Time

Box’s test of equality of covariance metrices found statistically significant differences (*p* = 0.000), and thus analyzed the ratio of G2/G1. The ratio was less than 1.5, so there was not any violation of homogeneity. Mauchly’s test for the sphericity analysis, in Skills, Kick, and Sum Time, was less than 0.75 (*p* < 0.05). Thus, to avoid any violation of sphericity, we checked Greenhouse Geisser. The degrees of Freedom were modified for Skills Time (*F* (1.165, 24.486) = 3.695, *p* = 0.061), Kicks Time (*F* (1.391, 29.218) = 3.862, *p* = 0.046), and Sum Time (*F* (1.266, 26.580) = 4.391, *p* = 0.038).

In between subject effects, statistically significant difference interactions were found in the Skills Time between groups in (F (1,21) = 9.720, *p* = 0.005, *η*^2^ = 0.316). Thus, Syntax analysis and, specifically, Bonferroni pairwise comparisons were utilized to find the differences between groups. The difference observed at the first measurement (40.4 ± 16.5 vs. 26.0 ± 5.3 sec, (95% CI (3.925, 24.802)), *p* = 0.009). Moreover, a statistically significant difference was found overall between measurements (*p* < 0.001) (Table 5).

### 3.4. Saliva Cortisol Concentration 

Box’s test of equality of covariance metrices had no statistically significant difference (*p* = 0.067); moreover, Levene’s test observed the *p*–value to be greater than 0.05, suggesting that there was not any violation in homogeneity. Then, an interaction between groups was found in the Multivariate test (Wilk’s lambda = 0.599, F (2.000, 20.000) = 6.696, *p* = 0.006). In pairwise comparison, a statistically significant difference in the second measurement was found between groups (CG vs. AG: 0.058 ± 0.12 vs. 0.122 ± 0.12 μg/dl, (95% CI (−0.99, −0.28), *p* = 0.001). Moreover, statistically significant decrement in saliva cortisol concentration was observed between the two measurements, (*p* < 0.001) (Figure 2).

### 3.5. Tec Pa’s Backstroke and Breststroke Scores

Box’s test of equality of covariance metrices found statistically significant differences (*p* = 0.044), thus analyzed the Ratio of G2/G1. The ratio was less than 1.5 so there was not violation of homogeneity. Using Mauchly’s test for Sphericity analysis, in BK and BR, ε was close to 1 (*p* > 0.05). Thus, checked Sphericity assumed for measurement (BK: F (2,42) = 51.388, *p* = 0.000 and BR: F (2,42) = 15.995, *p* = 0.000) and for measurement x group in which there was not any interaction (BK: F (2,42) = 2.491, *p* = 0.095 and BR: F (2,42) = 2.245, *p* = 0.118). 

In between subject effects, statistically significant differences were found between groups in both of BK (*p* = 0.009, *η*^2^ = 0.283) and BR (*p* = 0.020, *η*^2^ = 0.231). Thus, Syntax analysis and specifically Bonferroni pairwise comparisons were utilized to find the differences between groups. In BK, we observed a statistically significant difference in the third measurement (CG vs. AG: 6.6 ± 1.6 vs. 9.8 ± 1.4, 95% CI (−4.475, −1.931), *p* = 0.000), whereas a difference was observed in BR in the second (CG vs. AG: 1.3 ± 1.6 vs. 3.1 ± 2.2, 95% CI (−3.495, −0.126), *p* = 0.036) and third (CG vs. AG: 1.9 ± 2.5 vs. 4.5 ± 3.1, 95% CI (−5.031, −0.151), *p* = 0.038) measurement. Moreover, statistically significant differences were found between measurements in each group (*p* < 0.001) (Figure 3).

## 4. Discussion

The main objective of the study was to discover if an alternative swimming learning approach could give beneficial results on primary school-aged novice swimmers. Often, swimming coaches choose to teach young swimmers a variety of standardized swimming exercises which increase the boredom and restricts learning effectiveness. On the other hand, the main findings of the study indicate that a swimming learning approach which targets creative games and fun could make the swimming training more interesting and effective on skills and technique learning.

### 4.1. Movement Perception for Faster Skills’ and Technique’s Learning

The present study was conducted by the participation of young swimmers (7–9 years old). The aim of it was the children to learn the skills of start and dive, the techniques of BK and BR, and to improve their performance. The swimmers were chosen at these ages because according to the literature it is the most crucial age for faster learning [21]. Additionally, an additional feature of the sample’s selection was that at those ages they usually start swimming, and the main content of the courses is the specific skills and styles.

The most common learning approach which is used is through standardized exercises that are constantly repeated and prevent children from developing their imagination and ideas. In recent years, there has been a tendency to use a more playful approach. Many studies examined the effectiveness of that kind of training in sports such as track and field, tennis, football, and general skills, through the use of movements that children use in their daily routine [1,2,4,5,6,7,8].

The present study was based on the findings of the above research, comparing a standardized learning training approach with an alternative that targets mimicry and moving patterns which are usually used in a daily situation. Study results confirm that an alternative learning approach gives better results [2]. No similar research was found in swimming to compare the data. Only Magias and Pill [9] used a more distinct approach which helped the swimmers to develop the perception of the forces exerted on the body from the water.

The target in AG was to limit the faults that are usually observed at a move during an exercise and to give the children a better perception of their body movements. In the existing literature, athletes’ ability to perceive their movement was not mentioned. In the study, an important learning factor is movements perception. Generally, each child needs a different learning approach. However, common evidence on a learning process is the moves perception. Through the perception of an error, it is possible to learn the technique faster.

### 4.2. Technique, Skills and Performance 

In our study, the alternative exercises provided the opportunity for faster and more effective skills and technique assimilation. The playful spirit and content of the lessons created a positive learning background for both BK and BR styles. Additionally, the liquid element and the forces produced in the water did not seem to affect the children’s learning ability. Instead, they were found to have a better understanding of their body’s moves. This finding is in accordance with Magias and Pill’s [9] study results.

On performance variables (Skills’ time, Kicks’ time, and Sum time) in both groups were observed statistically significant improvements. Probably, the stimuli of the two training protocols have the same improvement in endurance. However, on AG compared to the CG less training volume (meters) was used because the emphasis was on the quality of movements’ execution and not on the quantity.

Additionally, for AG, those workouts, which contained more training volume (m) than the others, were performed in different directions inside the pool, in contrast to the CG that followed the usual route from one side of the pool to the other within the pool’s lane. The target was to differentiate the way that endurance is trained, taking the idea from the “IAAF Kids” [2] which used road exercises of various directions and obstacles.

In the studies of Papadimitriou and Savvoulidis [22,23], it was stated that endurance is a parameter that can be improved in childhood with a variety of training stimuli. In the present study, the training target was to learn the technique of BK and BR, but also, was to improve endurance. Since the endurance improved in both groups by performing technical exercises, it is understandable that with less fatigue in training the children can be improved in endurance.

### 4.3. Salivary Cortisol Concertation

Salivary cortisol concentration was used to examine children’s exercise stress levels [24]. Researchers use this index to understand the stress levels in acute or long time periods [25]. According to the literature, salivary cortisol concertation in both infants [26] and children [27] gave reliable results and showed that children’s mood depends on it and when they are engaged with one activity the cortisol’s concentration values alter.

The present study was based on the research of Sims et al., [10] who examined the effect of service quality in three daycare centers in relation to salivary cortisol concentration, based on samples taken from children. The results of the research showed that the children with the highest quality of service at daycare centers had the lowest salivary cortisol values.

In the study, it was found that the salivary cortisol concentration had no statistically significant difference at the first measurement. Therefore, the 45’ between groups possibly do not affect the salivary cortisol concentration. According to the literature, cortisol reached a peak at around 08:30am, then cortisol levels slowly decrease until the completion of the 24 h cycle [28]. Moreover, the circadian rhythm of each person differs, because cortisol’s Acrophase values vary from 07:59 to 09:05 am [28]. In contrast, statistically significant decrease was observed from the pre to post training measurements in both groups. The result is consistent with the literature about the effect of circadian rhythm, low intensity exercise, and the mood of participation on cortisol’s concentration reduction [10,11,28,29].

Another statistically significant difference which was observed in the second measurement between the groups, indicated that in CG, the cortisol concentration reduced more than in AG. Additionally, another difference is the rate of cortisol’s reduction during the respective hours that our measurements were performed. It was found that on children, the cortisol concentration from 09:30 to 10:15 am reduced by 10.5%, while the AG who trained at the same hours the reduction rate was 31.4%. Continuing from 10:15 to 11:00 am, the reduction rate on children is 14.7%, while the CG’s reduction rate was 48.7% [30,31].

The difference between groups is probably due to the fact that at the time of the day the CG participated, the environmental conditions were more delightful because the sunshine was more intense and brighter than AG’s hour. Therefore, there is a greater proportionate decrease in cortisol concentration at the same hours with an increased rate, possibly due to the circadian rhythm, low intensity exercise, and increased mood. However, further studies, at these ages are necessary.

### 4.4. Fear as Attenuate Factor on Skills and Technique’s Learning 

According to the results of the study, there was a greater improvement for AG at the skill of “Start” and at the BK and BR technique. In “Sink 1 and 2” similar improvement in both groups was observed. However, these two skills are very difficult for children to assimilate, especially when they are novices, due to the fear that exists when they immerse their head in the water.

Additionally, another measurement which was found statistically different between groups was in the first measurement of Skill’s time. That difference was observed because of the fear that the children felt in the first measurement, mainly in the CG. Thus, they needed more time to think how to perform each skill.

Usually, fear resulted from a previous traumatic experience or an attempt to protect themselves from an injury [32]. It is observed that children, when entering the water, choose to enter on foot or hold their nose to dive. These reactions are observed due to the safety that children feel when they step on their feet and because of the protection of their nose and mouth from the water’s possible entry. Therefore, in the study, fear was a reaction that was observed during the tests and training. After a short time period, the methodical teaching in both groups was contributed to overcoming the children’s fear. However, AG overcame this fear faster than CG, because the training’s content was focused on fun and recreational thinking.

According to the study results, the alternative swimming learning approach could be used in whole or as part of a training session. Children seem to prefer participating in a training session which gives the opportunity to think creatively and to learn, without recurring exhausting exercises. To make this happen, it is important the swimming coaches have the mood to create alternative solutions which will solve children’s motor learning difficulties faster than the usual standardized exercises that mostly utilize.

## 5. Conclusions

The alternative swimming training program was found to be more efficient or equally effective, compared to the standardized method, for teaching the skills of “Start” and “Sink”, at the improvement of BK and BR technique, at performance, and in the reduction of salivary cortisol concentration. Therefore, it is helpful for coaches to steer their swimming learning programs to the alternative form to achieve faster and more effective learning outcomes.

## 6. Study Limitations

In the present study, the swimmers’ age that was used in the intervention did not meet in any other study. Thus, the evaluation of skills and performance variables in those ages is novel. Despite this, there are some limitations in the study that probably affect the results of the study (Table 6).

## Figures and Tables

**Figure 1 healthcare-09-01234-f001:**
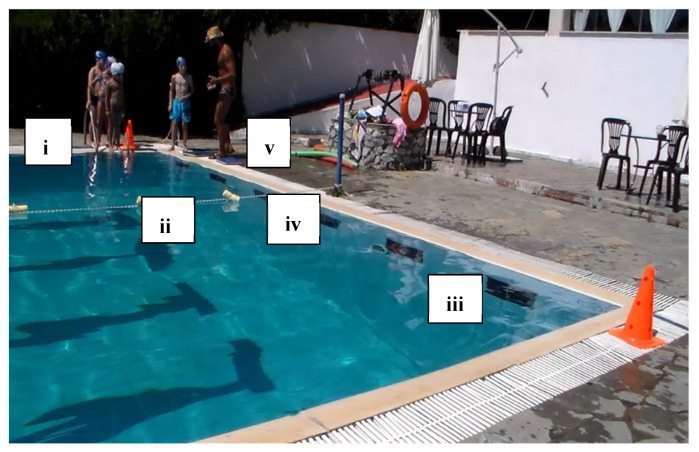
Skills and Performance test. **i** = Start: Entrance in the pool, **ii** = Sink 1: Dive under the pool’s lane for first time, **iii** = Swim: Swim until the opposite side of the pool, **iv** = Sink 2: Dive under the pool’s lane for second time, **v** = Kicks’ Time: Took the kickboard for the 35 m of free kick.

**Figure 2 healthcare-09-01234-f002:**
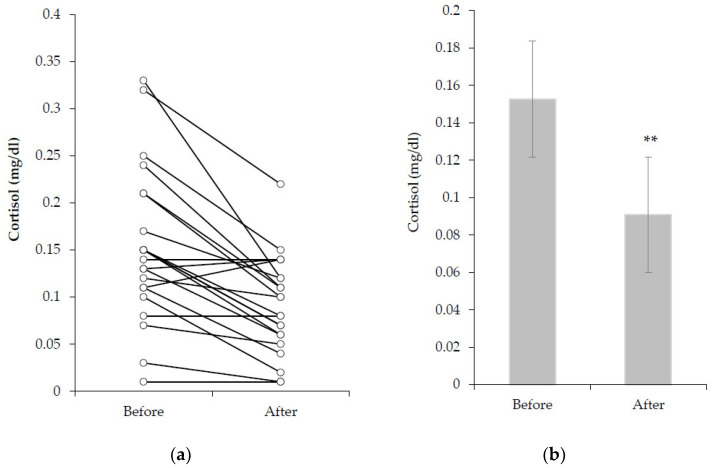
Saliva cortisol concentration before and after training, (**a**): for each swimmer, (**b**): mean values between groups. ** = Statistically significant difference between measurements.

**Figure 3 healthcare-09-01234-f003:**
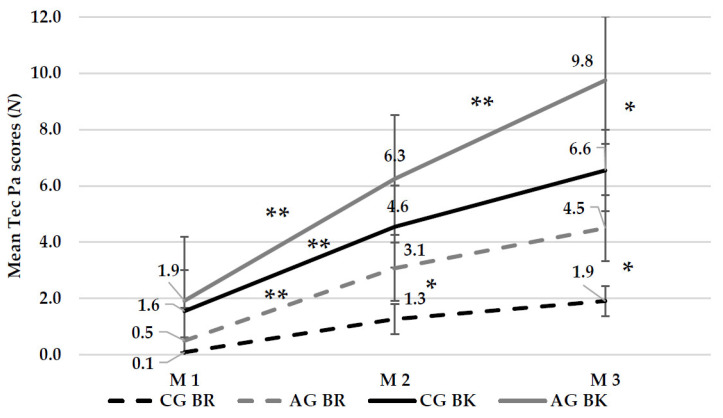
Swimmers’ mean score in Backstroke and Breaststroke. M 1 = Measurement 1, M 2 = Measurement 2, M 3 = Measurement 3, CG = Control Group, AG = Alternative Group, BR = Breaststroke, BK = Backstroke, * = Statistically significant difference between groups, ** = Statistically significant difference between measurements in each group.

**Table 1 healthcare-09-01234-t001:** Swimmers’ anthropometrics and intervention details.

Groups	Age (y)	Height (m)	Weight (kg)	Tanner (*n*)	Training Age (y)	Attendances (*n*)	Training Volume (m)
**CG**	9.1 ± 0.8	1.4 ± 0.1	33.8 ± 8.1	1.5 ± 0.7	2.2 ± 1.1	16.3 ± 2.1	400 ± 100
**AG**	9.0 ± 0.9	1.4 ± 0.1	33.3 ± 5.7	1.5 ± 0.5	3.0 ± 1.9	16.1 ± 2.8	200 ± 100

**CG** = Control Group, **AG** = Alternative Group.

**Table 2 healthcare-09-01234-t002:** Exercises between CG and AG.

CG	AG	Learning Target
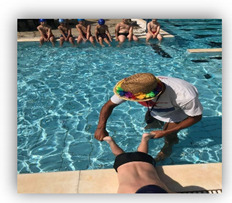	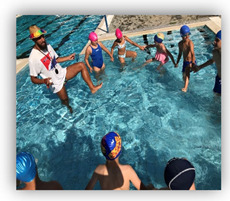	**Breaststroke kick** **CG**: Passive repetition **AG**: Dancing
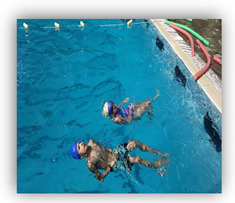	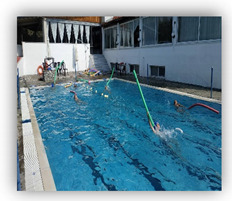	**Backstroke body position** **CG**: Star position **AG**: Tube for pillow
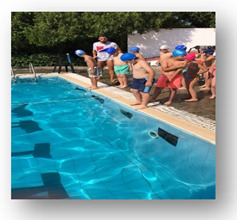	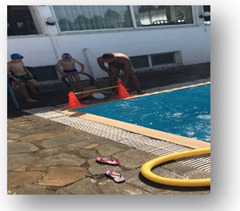	**Start** **CG**: Entrance from the pool’s wall **AG**: Soldier’s entry
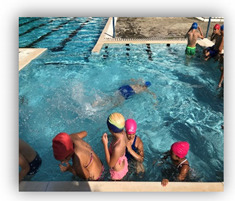	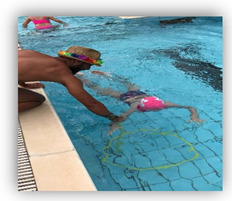	**Sink** **CG**: Diving try in small pool **AG**: Ring door

**Table 3 healthcare-09-01234-t003:** Measurements’ schedule.

0 Weeks	4 Weeks	8 Weeks
Technique	Technique	Technique
Skills and Performance	Skills and Performance	Skills and Performance
	Saliva cortisol concentration (Before and After training)	

**Table 4 healthcare-09-01234-t004:** Percentage of successful tries within groups in the three measurements.

Start (N%)					Sink 1 (N%)				Sink 2 (N%)			
M	CG	AG	Fisher’s (*p*–Value)	Cochran’s (*p*–Value)	CG	AG	Fisher’s (*p*–Value)	Cochran’s (*p*–Value)	CG	AG	Fisher’s (*p*–Value)	Cochran’s (*p*–Value)
**M 1**	9.1	58.3	0.027 *	0.001 **	36.4	50	0.680	0.003 **	45.5	66.7	0.414	0.043 **
**M 2**	36.4	66.7	0.220		90.9	83.3	1		45.5	75	0.214	
**M 3**	63.6	100	0.037 *		90.9	75.0	0.590		100	75	0.217	

M = Measurement, M 1 = Measurement 1, M 2 = Measurement 2, M 3 = Measurement 3, CG = Control Group, AG = Alternative Group, * = Statistically significant difference within groups (Fisher’s, *p*–value), ** = Statistically significant difference between measurements (Cochran’s, *p*–value).

**Table 5 healthcare-09-01234-t005:** Performance variables within groups in the three measurements.

Skills Time (sec)				Kicks Time (sec)			Sum Time (sec)		
M	CG	AG	Eta (*η*^2^)	CG	AG	Eta (*η*^2^)	CG	AG	Eta (*η*^2^)
**M 1**	40.4 ± 16.5 *	26.0 ± 5.3	0.316	112.9 ± 34.7 **	93.8 ± 24.7	0.035	153.8 ± 49.5 **	120.4 ± 27.0	0.101
**M 2**	30.4 ± 7.7 **	26.3 ± 5.7		77.8 ± 11.2 **	80.8 ± 21.4		108.0 ± 11.9 **	107.3 ± 26.3	
**M 3**	26.8 ± 5.2 **	23.2 ± 3.8		77.2 ± 12.8	72.7 ± 17.4		105.1 ± 13.3	95.8 ± 20.8	

M = Measurement, M 1 = Measurement 1, M 2 = Measurement 2, M 3 = Measurement 3, CG = Control Group, AG = Alternative Group, * = Statistically significant difference between groups, ** = Statistically significant difference between measurements.

**Table 6 healthcare-09-01234-t006:** Study’s limitations.

Limitation	Problem	Future Solution
The weather conditions were unstable.	Limited the presence of children in the training.	To take part in an indoor pool.
Due to illness, the presence of some children was small.	Smaller samples were used for the statistical analysis.	Smaller intervention periods.
The start time of training differed between the two groups by 45 min.	The AG had earlier training than CG, thus there were complaints from the children of AG.	The training must be starting at the same time, probably on different days.

## Data Availability

Not applicable.

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
