# Peer review of "The Effect of an Alternative Swimming Learning Program on Skills, Technique, Performance, and Salivary Cortisol Concentration at Primary School Ages Novice Swimmers"

_healthcare, 2021, doi:10.3390/healthcare9091234_

Round 1

Reviewer 1 Report

The aim of the present study was to evaluate the efficiency of an alternative swimming learning approach on several swimming parameters in children. Although the general concept is towards the correct direction which is to make children happy while they are exercising/training, there are several issues throughout the manuscript that should be assessed by the authors.

  • The authors are suggested to check the entire manuscript for writing clarity and grammatical errors. Indicatively, among many others, many sentences are vague and should be re-phrased in a more understandable manner (lines 12-14, 24-26, 29-30, 45-47, 57-59, 120-121, 152-157, 295-299....)
  • Line 90: it is suggested to explain to the readers what is Tanner stage and how this can affect the selection of groups. 
  • lines 95-96: please include the information that a consent form was signed by their legal guardian and in the methodology section as well.
  • Why freestyle was not evaluated which is the most common style that children first learn? The explanation of enthusiasm and boredom (lines 110-111) is arbitrary.
  • Since "TecPa" was the main method for evaluating skills and technique, the authors are suggested to explain how this is happening and provide details about that method.  
  • One of the biggest concerns is that  the evaluation of swimmers performance was made by only one swimming coach. How the authors were assured that the results were fully impartial? This is something that should also be mentioned among with the other limitations at least.
  • Figure 1. Apart from the individual data it will help the reader the presentation of the mean values as two columns for example.
  • The results from salivary cortisol measurements are confusing. The different time of training is a serious confounding factor that weakens the methodology of the study. It is well documented that cortisol follows circadian rhythms...
  • Line 369: this research had no funds!

Author Response

We want to thank the reviewer for the really valuable comments which increase the article’s status. We believe that the answers and corrections will satisfy the reviewer.

  • The authors are suggested to check the entire manuscript for writing clarity and grammatical errors. Indicatively, among many others, many sentences are vague and should be re-phrased in a more understandable manner (lines 12-14, 24-26, 29-30, 45-47, 57-59, 120-121, 152-157, 295-299....)

Answer

The whole manuscript was checked for writing clarity and grammatical errors.

  • Line 90: it is suggested to explain to the readers what is Tanner’s stage and how this can affect the selection of groups. 

Answer

Explained what is Tanner’s stage and how this affected the selection of the groups. Lines: 579 – 582.

  • lines 95-96: please include the information that a consent form was signed by their legal guardian and in the methodology section as well.

Answer

The information was included. Lines: 588 – 589.

  • Why freestyle was not evaluated which is the most common style that children first learn? The explanation of enthusiasm and boredom (lines 110-111) is arbitrary.

Answer

It is true that freestyle is the most common style which children learn. However, backstroke and breaststroke styles were chosen because of the difference in the move symmetry, the multiple muscle activation, and the novelty of their use instead of freestyle that usually is used in studies. Lines: 702 – 704.

  • Since "Tec Pa" was the main method for evaluating skills and technique, the authors are suggested to explain how this is happening and provide details about that method.

Answer

Tec Pa was the tool which swimmers were evaluated in backstroke and breaststroke styles. More details were added for better methods understanding. Lines: 801 – 810.

  • One of the biggest concerns is that the evaluation of swimmers’ performance was made by only one swimming coach. How the authors were assured that the results were fully impartial? This is something that should also be mentioned among the other limitations at least.

Answer

I understand your concern. However, the experienced coach who was responsible for swimmers’ evaluation did all three measurements, thus there is not any bias about evaluation’s reliability. Moreover, Tec Pa is already reliable and valid for use [1]. Also, under real conditions, every swimming coach evaluates their swimmers. However, in our study, we want to make blind measurements. That’s why we use different swimming coaches for swimmers’ evaluation, for the planning of training sessions, and for their execution. Lines: 801 – 810. 

[1] Papadimitriou, K.; Papadimitriou, N.; Gourgoulis, V.; Barkoukis, V.; Loupos, D. Assessment of Young Swimmers’ Technique with Tec Pa Tool. CEJSSM, 2021, 34, 39–51.

  • Figure 1. Apart from the individual data, it will help the reader the presentation of the mean values as two columns for example.

Answer

A new figure was added to present the mean values. Line: 983.

  • The results from salivary cortisol measurements are confusing. The different time of training is a serious confounding factor that weakens the methodology of the study. It is well documented that cortisol follows circadian rhythms.

Answer

We discuss extensively that the 45’ do not affect the circadian rhythm [1 – 3]. Lines: 1119 – 1195.

  1. S; Debono, M. Replication of cortisol circadian rhythm: new advances in hydrocortisone replacement therapy. Ther Adv Endocrinol Metab. 2010, 1(3), 129-138.
  2. Hindmarsh, P.C.; Geertsma, K. Congenital Adrenal Hyperplasia. A Comprehensive guide. Academic Press, 2017; pp. 10.
  3. Bartels, Μ.; de Geus, E.J.C.; Kirschbaum, C.; Sluyter, F.; Boomsma D.I. Heritability of Daytime Cortisol Levels in Children. Beh. Gen. 2003, 33, 421–433.
  • Line 369: this research had no funds!

Answer

Was corrected. Line: 1288.

Reviewer 2 Report

ABSTRACT

Lines 12-14: add the eligibility criteria and the experience level of participants

Line 14: add some information about the experimental approach (number of weeks, training frequency) and the difference between alternative and classic approach

Line 14-18: it would be nice to reorganize the results in “between” and “within”-groups results.

INTRODUCTION

Line 24: describe in detail the meaning of “playful training” namely the type of training approach, the background and the type of exercises and differences with other approaches

Lines 25-27: in comparison to what?

Line 29: support the assumption with a stronger experimental basis and references.

Line 37: Add the reference

Lines 31-82: the rationale can be improved. Now, each paragraph seems to be a presentation of different studies not connected in a rationale. Maybe is better to go through the introduction by justifying the causes for the findings and presenting the main outcomes and interactions with the training process, justifying the use of them in the current article.

Line 83: the objective is presented without a statement of contribution. Before the objective, it is determinant to add a statement of contribution and motivation for the study.

METHODS

Line 89: add the a priori sample size calculation

Line 91: Give the eligibility criteria and the sources and methods of selection of

participants. Describe methods of follow-up

Lines 93-94: randomization was made? Describe the methods of randomization. Also, add information about differences or not between baseline levels of the groups.

Line 101: add a brief sentence with the type of study design.

Line 105: Describe the setting, locations, and relevant dates, including periods of recruitment, exposure, follow-up, and data collection

Line 111: describe who acted as coach, who made the observations (assessments), and who made the statistics. If done, who was blinded after assignment to interventions (for example, participants, care providers, those assessing outcomes) and how?

Line 114: my recommendation is to add an appendix with examples of training plans for one week in both groups helping the readers to understand the exercises and approaches.

Line 141: add the reliability level of the coaches to assess this. This can be a high source of bias.

Lines 139-172: generally, the authors must follow this recommendation: “For each variable of interest, give sources of data and details of methods of assessment (measurement). Describe comparability of assessment methods if there is

more than one group” and “Explain how quantitative variables were handled in the analyses. If applicable, describe which groupings were chosen and why”

Line 180: add the values of normality and homogeneity of each outcome.

RESULTS

Figure 1. My suggestion is to make a new figure following figure 2 of this article: Nimphius, S., & Jordan, M. J. (2020). Show Me the Data, Jerry! Data Visualization and Transparency. International Journal of Sports Physiology and Performance15(10), 1353-1355.

DISCUSSION

Start the first paragraph with the objective answer to the main objectives of the study.

Add a paragraph of practical implications at the bottom of the discussion

Author Response

We want to thank the reviewer for the really valuable comments which increase the article’s status. We believe that the answers and corrections will satisfy the reviewer.

  • Lines 12-14: add the eligibility criteria and the experience level of participants
  • Line 14: add some information about the experimental approach (number of weeks, training frequency) and the difference between alternative and classic approach.
  • Line 14-18: it would be nice to reorganize the results in “between” and “within”-groups

Answer

Some of the eligibility criteria, the experience and level of swimmers, and some information about intervention were added. However, according to Journal’s “Instructions for authors,” the total number of words must be a maximum of 200 words. Thus, we kept some of our specific information in “Methodology” and “Results” to maintain these words limit. Lines: 9 – 21.

Here are the instructions

Abstract: The abstract should be a total of about 200 words maximum. The abstract should be a single paragraph and should follow the style of structured abstracts, but without headings: 1) Background: Place the question addressed in a broad context and highlight the purpose of the study; 2) Methods: Describe briefly the main methods or treatments applied. Include any relevant preregistration numbers, and species and strains of any animals used. 3) Results: Summarize the article's main findings; and 4) Conclusion: Indicate the main conclusions or interpretations. The abstract should be an objective representation of the article: it must not contain results which are not presented and substantiated in the main text and should not exaggerate the main conclusions.

INTRODUCTION

  • Line 24: describe in detail the meaning of “playful training” namely the type of training approach, the background and the type of exercises, and differences with other approaches.

Answer

Described in detail the meaning of “playful training”, the background, the type of exercises, and differences with other approaches.

which in the literature has the same   After the reviewer’s comments on the first paragraph added: the meaning of “playful training, the background, the type of exercises and differences with other approaches and the comparison. Lines: 25 – 35.

  • Lines 25-27: in comparison to what? 

Answer

In comparison with a classic training method. Line: 35.

  • Line 29: support the assumption with a stronger experimental basis and references.

Answer

Added more information supporting by the same reference. The experimental basis is described more extend from the next paragraph. Lines: 33 – 35.

  • Line 37: Add the reference

Answer

The reference was added. Line: 37.

  • Lines 31-82: the rationale can be improved. Now, each paragraph seems to be a presentation of different studies not connected in a rationale. Maybe is better to go through the introduction by justifying the causes for the findings and presenting the main outcomes and interactions with the training process, justifying the use of them in the current article.

Answer

Now, the introduction’s paragraphs are connected. However, with all of the respect to the reviewer, our main target in the introduction is to show the existing literature and no to justify it. Thus, we keep the literature’s justification in the “Discussion”. Lines: 36 – 554.

  • Line 83: the objective is presented without a statement of contribution. Before the objective, it is determinant to add a statement of contribution and motivation for the study.

Answer

A statement of contribution and motivation for the study was added. Lines: 549 – 554.

METHODS

  • Line 89: add the a priori sample size calculation

Answer

Priori sample size calculation was added. Lines: 557 – 558.

  • Line 91: Give the eligibility criteria and the sources and methods of selection of participants. Describe methods of follow-up

Answer

Eligibility criteria and the sources and methods of selection of participants were added. Lines:558 – 563.

  • Lines 93-94: randomization was made? Describe the methods of randomization. Also, add information about differences or not between baseline levels of the groups.

Answer

We describe the randomization, the methods of it, and information about the non-differences between baseline levels of the groups. Lines: 572 – 574.

  • Line 101: add a brief sentence with the type of study design.

Answer

A brief sentence was added. Lines: 601 – 602.

  • Line 105: Describe the setting, locations, and relevant dates, including periods of recruitment, exposure, follow-up, and data collection.

Answer

More details were added in the “Intervention’s details” and “Measurements” sections. Lines: 603 - 610.

  • Line 111: describe who acted as coach, who made the observations (assessments), and who made the statistics. If done, who was blinded after assignment to interventions (for example, participants, care providers, those assessing outcomes) and how?

Answer

Information was added. Lines: 718 – 722.

  • Line 114: my recommendation is to add an appendix with examples of training plans for one week in both groups helping the readers to understand the exercises and approaches.

Answer

Appendix with examples of training plans was added at the end of the manuscript. Line: 1394.

  • Line 141: add the reliability level of the coaches to assess this. This can be a high source of bias.

Answer

I understand your concern. However, the experienced coach who was responsible for swimmers’ evaluation did all three measurements, thus, there is not any bias about evaluation reliability. Moreover, Tec Pa is already reliable and valid for use [1]. Also, under real conditions, every swimming coach evaluates their swimmers. However, in our study, we want to make blind measurements. That’s why we use different swimming coaches for swimmers’ evaluation, for the planning of training sessions, and for their execution. Lines: 801 – 810. 

[1] Papadimitriou, K.; Papadimitriou, N.; Gourgoulis, V.; Barkoukis, V.; Loupos, D. Assessment of Young Swimmers’ Technique with Tec Pa Tool. CEJSSM, 2021, 34, 39–51.

  • Lines 139-172: generally, the authors must follow this recommendation: “For each variable of interest, give sources of data and details of methods of assessment (measurement). Describe comparability of assessment methods if there is more than one group” and “Explain how quantitative variables were handled in the analyses. If applicable, describe which groupings were chosen and why”

Answer

More details for the methods were given. Lines: 641 – 729.

  • Line 180: add the values of normality and homogeneity of each outcome.

Answer

The values of normality and homogeneity of each outcome were added. Line: 730.

RESULTS

  • Figure 1. My suggestion is to make a new figure following figure 2 of this article: Nimphius, S., & Jordan, M. J. (2020). Show Me the Data, Jerry! Data Visualization and Transparency. International Journal of Sports Physiology and Performance15(10), 1353-1355.

Answer

New figures following “Figure 2” were added. Line 796.

DISCUSSION

  • Start the first paragraph with the objective answer to the main objectives of the study.

Answer

Added the first paragraph with the objective answer to the main objectives of the study. Lines: 1035 – 1041.

  • Add a paragraph of practical implications at the bottom of the discussion

Answer

A paragraph of practical implications was added. Lines: 1215 – 1220.

Round 2

Reviewer 1 Report

Thew authors have significantly ameliorated their manuscript. My comments were answered efficiently. One question though: Table 1: what Adherence stands for?

Author Response

The authors have significantly ameliorated their manuscript. My comments were answered efficiently. One question though: Table 1: what Adherence stands for?

  • It is "attendances" now corrected. Line: 131

Reviewer 2 Report

The article was improved based on the comments

Author Response

There are no comments.